# Genetic Diversity and Phenotypic Variation of Indigenous Wild Cherry Species in Kazakhstan and Uzbekistan

**DOI:** 10.3390/plants14111676

**Published:** 2025-05-30

**Authors:** Ulzhan Manapkanova, Nazgul Rymkhanova, Stefanie Reim, Eric Fritzsche, Monika Höfer, Natalya Beshko, Yeskendir Satekov, Svetlana V. Kushnarenko

**Affiliations:** 1Institute of Plant Biology and Biotechnology, 45 Timiryazev Str., Almaty 050040, Kazakhstan; n.rymkhanova@gmail.com; 2Faculty of Biology and Biotechnology, Al-Farabi Kazakh National University, 71 Al-Farabi Av., Almaty 050040, Kazakhstan; 3Julius Kühn-Institute (JKI), Federal Research Centre for Cultivated Plants, Institute for Breeding Research on Fruit Crops, 3a Pillnitzer Platz, 01326 Dresden, Germany; stefanie.reim@julius-kuehn.de (S.R.); eric.fritzsche@julius-kuehn.de (E.F.); monika.hoefer@julius-kuehn.de (M.H.); 4Institute of Botany, 32 Durmon Yuli Str., Tashkent 100125, Uzbekistan; natalia.beshko@gmail.com; 5Institute of Botany and Phytointroduction, 36 Timiryazev Str., Almaty 050040, Kazakhstan; irokezz@inbox.ru

**Keywords:** *Prunus* spp., wild cherry species, genetic diversity, SSR markers, biodiversity

## Abstract

This study investigates the phenotypic characteristics, genetic diversity, and population structure of four wild cherry species collected from various regions of Kazakhstan and Uzbekistan: *Prunus fruticosa* Pall., *Ptunus erythrocarpa* (Nevski) Gilli, *Prunus griffithii* var. *tianshanica* (Pojark.) Ingram, and *Prunus verrucosa* (Franch.). A total of 163 accessions were characterized morphologically using standardized descriptors for plant, leaf, and fruit traits. Genetic diversity was assessed using 13 simple sequence repeat (SSR) markers. STRUCTURE analysis revealed that 87.7% of the accessions were assigned to pure species. However, hybrid accessions were identified in *P. griffithii* var. *tianshanica* (34.4%), *P. erythrocarpa* (18.5%), and *P. verrucosa* (8.0%). Identical genotypes were found across all species, with *P. fruticosa* showing the highest proportion (54.8%), likely due to clonal propagation via root suckers. Among the four species, *P. verrucosa* exhibited the highest genetic diversity, while *P. fruticosa* had the lowest. Analysis of molecular variance (AMOVA) showed that genetic variation within the species (81%) was substantially greater than variation among the species (19%). These findings enhance our understanding of the genetic relationships among wild cherry species in Central Asia and provide valuable data for conservation planning and breeding programs aimed at improving drought and frost tolerance in *Prunus* species.

## 1. Introduction

Cherries are one of the most popular stone fruit crops. Global cherry production reached approximately 2.77 million tons in 2022, marking a slight increase in production from the previous year [1]. Over 30 species of cherry have been identified worldwide, but only two species are primarily used in industrial production: *Prunus avium* (sweet cherry) and *Prunus cerasus* (sour cherry) [2]. Cherries are popular not only because of their excellent taste, but also because of their health benefits. Cherries’ high phenols content (anthocyanins and hydroxycinnamic acids) gives them antioxidant and anti-inflammatory properties [2]. Recent studies have shown promising results regarding the impact of sweet cherries on the regulation of cell proliferation and metabolic reprogramming of cancer cells. Therefore, cherries could be used as a dietary supplement in anti-cancer therapy [3].

Four wild cherry species grow in Kazakhstan: *Prunus fruticosa* Pall., *Prunus erythrocarpa* (Nevski) Gilli, *Prunus griffithii* var. *tianshanica* (Pojark.) Ingram, and *Prunus verrucosa* Franch [4,5]. These species grow in different climatic regions of Kazakhstan. *P. fruticosa* populations are found in the steppe and forest–steppe zones in central and northern Kazakhstan. This species is a low-growing, drought-resistant, xerophytic shrub with high winter hardiness. The other three species are also low shrubs, reaching 1 to 2 m in height, and are found on mountain slopes, primarily in the western Tien Shan [4,5,6]. Three species of wild cherry are also found in Uzbekistan: *P. erythrocarpa*, *P. griffithii* var. *tianshanica*, and *P. verrucosa* [7,8]. While *P. fruticosa* is classified in the subgenus Cerasus, section Eucerasus, the other three species are classified in the section Microcerasus [9,10].

The biodiversity of these wild cherry species provides an inexhaustible gene pool for breeding. The wild cherry species of Kazakhstan and Uzbekistan, in particular, are of interest as a source of genes for high drought and frost resistance and could be used to develop new winter-hardy and drought-resistant varieties, as well as dwarf rootstocks for cultivated cherry and plum varieties [4,5,8].

However, the effective use of this genetic potential in breeding programs requires a thorough understanding of the genetic diversity and structure of wild cherry populations. In this regard, molecular genetic tools play a crucial role. Microsatellite DNA markers (SSR markers) are currently widely used to study the genetic diversity of wild species and to identify varieties of agricultural crops, including cherries [11,12,13,14,15]. The European Collaborative Programme for Genetic Resources (ECPGR) has developed recommendations on the use of a set of 16 microsatellite (SSR) markers for identifying cultivars, hybrids, and wild cherry species [16].

Knowledge of population genetic diversity and genetic structure as well as the identification of hybrids and identical genotypes are crucial for implementing sustainable conservation strategies and for future breeding programs. For closely related species, it can be difficult to distinguish between them. The differentiation of *P. fruticosa, P. erythrocarpa, P. griffithii* var. *tianshanica*, and *P. verrucosa* is particularly challenging due to their close relationship. Although botanical descriptions of wild cherry species in Kazakhstan and Uzbekistan have been carried out in the past [4,5,6,7,8], a more detailed molecular genetic study is still lacking.

This research aimed to conduct a phenotypic and molecular genetic analysis of the biodiversity of four wild species in Kazakhstan and Uzbekistan using microsatellite markers. Molecular genetic methods, particularly the use of microsatellite (SSR) markers, could enable the reliable differentiation of morphologically similar wild cherry species that cannot be accurately distinguished based on phenotypic traits alone. This study represents the first comprehensive molecular and phenotypic analysis of these four species.

## 2. Results

### 2.1. Phenotypic Evaluation of Prunus Species from Kazakhstan and Uzbekistan

*P. griffithii* var. *tianshanica*—Forty-five accessions were collected from four populations in Kazakhstan (in the Almaty region, in the Big Aksu gorge, and in the Turkestan region in the Sairam-Ugam National Park) and from one population in Uzbekistan (in the Tashkent region in the Ugam-Chatkal National Park) (see Figure 1 and Figure 2 and Table 1). The places of growth are stony slopes and rocks in the foothills and the lower and middle belts of the mountains of the Northern (Almaty region) and Western Tien Shan (Turkestan and Tashkent regions). These areas are characterized by a continental climate with hot summers (average temperature, 36–38 °C) and cold winters (average temperature, −4 °C) [17].

The accessions were found at altitudes of 1124–1466 m a.s.l. in Kazakhstan, on rocky slopes dominated by shrub thickets consisting of *Rosa kokanica*, *Centaurea*, and *Bromus*. In Sairam-Ugam National Park, *P. griffithii* var. *tianshanica* shrubs grew alongside *P. verrucosa* and *P. erythrocarpa* within the same phytocenoses.

Accessions of *P. griffithii* var. *tianshanica* were collected in the territory of Ugam-Chatkal National Park in Uzbekistan, at an altitude of 1689–1753 m a.s.l. It grows alongside *P. erythrocarpa* as a co-dominant species within the grass–forb–shrub community (*Poa bulbosa* L., *Atraphaxis pyrifolia* Bunge, et al.), alongside juniper (*Juniperus seravschanica* Kom.) and honeysuckle (*Lonicera altmannii* Regel & Schmalh., *L. nummulariifolia* Jaub. & Spach).

*P. griffithii* var. *tianshanica* is a shrub with a height ranging from 39.3 to 119.4 cm and a mean height of 69.27 ± 33.31 cm, as determined in the studied populations. In the Almaty region, however, the shrubs are shorter, reaching heights of 15–90 cm, with an average height of 44.1 ± 24.56 cm. The leaves are ovate–lanceolate, with a pointed apex and a cuneate base. The average leaf area of *P. griffithii* var. *tianshanica* in the studied populations was 1.42 ± 0.43 cm^2^ (Figure 1D). The leaf margins are biserrate, with visible sharp teeth. Flowering occurs in April–May with pink flowers (Figure 1B), and fruiting takes place in July–August (Figure 1D). The fruits are small and round, with an average length of 0.68 ± 0.01 cm and a width of 0.65 ± 0.08 cm (Figure 1C,D). The taste of the fruit varies from sour to sour-sweet and sweet.

*P. erythrocarpa*—A total of 34 accessions of *P. erythrocarpa* were collected from two populations in Kazakhstan and three populations in Uzbekistan (Figure 1E–H and Figure 2 and Table 1). The shrubs grew on dry rocky slopes in the foothills and low and mid mountains of the Turkistan and Tashkent regions, at an altitude of 716–2069 m a.s.l. *P. erythrocarpa* shrubs were found alongside *P. griffithii* var. *tianshanica* and *P. verrucosa* in the Sayram-Ugam National Park and alongside *P. griffithii* var. *tianshanica* in the Ugam-Chatkal National Park. In the Jizzakh region of Uzbekistan (Pamir–Alay Mountains), one accession was collected alongside *P. verrucosa*.

The average height of the shrubs in the studied populations varied from 52.4 to 140 cm, with an average of 106 ± 39.57 cm. The leaves are elongated and oval with a pointed apex and a cuneate base (Figure 1H). The leaf margins are serrated or serratulate with small teeth. *P. erythrocarpa* is phenotypically very similar to *P. verrucosa* but can be distinguished by its white-tomentose abaxial leaf surface. The average leaf area in the studied populations was 1.74 ± 0.58 cm^2^. Flowering occurs in April–May with pink flowers (Figure 1F), and fruiting takes place in July–August. The fruits are small and round, with an average length of 0.7 ± 0.05 cm and a width of 0.77 ± 0.05 cm (Figure 1H). The taste of the fruit varies from sour to sour-sweet and sweet.

*P. verrucosa*—A total of 53 accessions of *P. verrucosa* were collected from three populations in Kazakhstan and two populations in Uzbekistan (Figure 1I–L and Figure 2 and Table 1). The shrubs grew on rocky and gravelly slopes in the foothill regions of the Turkistan, Tashkent, and Jizzakh regions at an altitude of 1409–1554 m a.s.l. *P. verrucosa* shrubs were found alongside *P. erythrocarpa* and *P. griffithii* var. *tianshanica* in the same locations in the Turkistan region. In the Jizzakh region of Uzbekistan (Pamir–Alay Mountains), thirteen accessions grew alongside *P. erythrocarpa*. The average height of the shrubs in the studied populations varied from 72.6 to 127.8 cm, with an average of 89.36 ± 22.11 cm. The species is distinguished by its brownish-gray branches covered with dry scales (warts). The leaves are oblong–elliptical or obovate, with a slightly pointed apex and a cuneate base. The leaf margins are serrated with sharp teeth. The average leaf area in the studied populations was 1.57 ± 0.54 cm^2^ (Figure 1L). The flowering period occurs in April–May, with pink flowers (Figure 1J), and fruiting takes place in June–July. The fruits are small and round with an average length of 0.73 ± 0.01 cm and a width of 0.79 ± 0.05 cm (Figure 1L). The fruit has a sour-sweet taste with a tart aftertaste.

*P. fruticosa*—A total of 31 accessions were collected from three populations in the Kostanay region (Figure 1M–P and Figure 2 and Table 1). This is a forest–steppe zone of Kazakhstan, characterized by high summer temperatures (average of 26 °C) and low winter temperatures (average of −16 °C). The accessions grew at altitudes of 153–202 m a.s.l., on the forest edges. The upper layer is dominated by *Betula pendula* Roth or *Pinus sylvestris* L., as well as associated shrub species such as *Rosa* sp. and *Spiraea hypericifolia*. The soil in the growing areas is sandy and/or clayey. The average shrub height in the studied population varied from 58.5 to 101.9 cm, with an average of 69.98 ± 23.86 cm. Compared to the other three species, *P. fruticosa* had the largest leaves, with an average leaf area of 4.7 ± 0.77 cm^2^, in the populations studied (Figure 1P). The leaves are elliptical, with the greatest width closer to the middle. Both the apex and the base are pointed. The leaf margins are weakly serrated. The flowering period occurs in April–May, with white flowers (Figure 1N), and fruiting takes place in July–August. Although the cherry plants studied in the Kostanay region were in relatively good condition, fruiting was absent, or only a few fruits were observed. The fruits are small, with an average length of 1.1 ± 0.11 cm and width of 0.93 ± 0.07 cm and have a pleasant sour-sweet taste (Figure 1O,P). Unlike the other species, *P. fruticosa* has a noticeable pedicel, with an average length of 2.8 ± 0.44 cm.

In the Sayram-Ugam National Park, three species—*P. griffithii* var. *tianshanica*, *P. verrucosa*, and *P. erythrocarpa*—were found in the same phytocenoses in the foothills of the Western Tien Shan at an altitude of 715–854 m a.s.l., growing in close proximity to each other. Sometimes, identifying the species based solely on morphological characteristics was challenging. For instance, *P. erythrocarpa* is distinguished by a white tomentum on the underside of its leaves, whereas *P. verrucosa* is characterized by dry scales (warts) on its trunks and branches. However, some accession has both features, making systematic classification difficult.

### 2.2. Genetic Diversity Parameters of the Prunus Species from Kazakhstan and Uzbekistan

To estimate the genetic diversity parameters of the four cherry species native to Kazakhstan and Uzbekistan, 163 accessions were analyzed based on their molecular characteristics. Analysis using 13 SSR markers revealed a high level of genetic diversity within the cherry species studied.

Initial identification based on morphological characteristics sometimes did not allow for a clear differentiation between species. Therefore, the model-based clustering method implemented in STRUCTURE revealed several inconsistencies between the morphological and genetic classifications. Specifically, four accessions initially described as *P. griffithii* var. *tianshanica* were reclassified based on SSR data, which resulted in three of them being genetically assigned to *P. verrucosa*, and one to *P. erythrocarpa*. This highlights the limitations of relying solely on morphology for species identification. Of the 163 accessions analyzed, 143 (87.7%) were assigned to distinct genetic clusters as pure accessions. Consequently, these 143 accessions were used for further analysis and included 33 accessions of *P. griffithii* var. *tianshanica*, 29 accessions of *P. erythrocarpa*, 31 accessions of *P. fruticosa*, and 51 *P. verrucosa* accessions. The remaining 20 accessions were identified as hybrids and were excluded from subsequent genetic analysis. The analysis revealed the presence of hybrids in three species (*P. griffithii* var. *tianshanica*, *P. erythrocarpa*, and *P. verrucosa*) but not in *P. fruticosa*. The highest percentage was found in *P. griffithii* var. *tianshanica* (34.4%), followed by *P. erythrocarpa* (18.5%) and *P. verrucosa* (8.0%).

After the hybrids were excluded, an initial analysis was performed using the GenAlEx software package and143 pure accessions. Firstly, to ensure the accuracy of the subsequent analysis, the ‘multilocus match analysis’ tool for codominant data was used to identify genetically identical genotypes. Of the four cherry species studied, 26 accessions were found to be genetically identical. For subsequent analyses, only one representative of each unique genetic profile was included (six out of 26), while the remaining duplicates were excluded. Identical genotypes were found in all four studied species, with the largest proportion (54.8% of the total number of accessions collected) occurring in *P. fruticosa*. Further analysis was performed on 123 accessions.

The genetic diversity parameters of the studied *Prunus* species studied are shown in Table 2. The analysis revealed variation in the number of alleles (Na) and the number of effective alleles (Ne) among the species. *P. verrucosa* had the highest average number of alleles (Na = 14.04), while *P. fruticosa* had the lowest (Na = 2.19). The number of effective alleles (Ne) ranged from 1.71 (*P*. *fruticosa*) to 5.82 (*P. erythrocarpa*), indicating a range of allelic diversity that varies between the species studied.

The values of the Shannon index (I), which reflects the level of genetic diversity, ranged from a low of 0.54 for *P. fruticosa* to a high of 2.10 for *P. verrucosa*. Observed heterozygosity (Ho) was highest in *P. verrucosa* (0.57) and lowest in *P. fruticosa* (0.42), indicating differences in heterozygosity between the species. Expected heterozygosity (He) showed a similar trend, ranging from 0.30 in *P. fruticosa* to 0.80 in *P. verrucosa*. Unbiased expected heterozygosity (uHe) was also highest in *P. verrucosa* (0.84) and lowest in *P. fruticosa* (0.32), further confirming the differences in the level of genetic diversity. The fixation coefficient (F), which reflects the level of inbreeding, ranged from a negative value of −0.41 for *P. fruticosa* (indicating the presence of an excess of heterozygotes) to a value of 0.27 for *P. verrucosa* (indicating a moderate level of inbreeding).

The obtained results demonstrate remarkable differences in genetic diversity between the studied *Prunus* species. *P. verrucosa* exhibited the highest genetic diversity, while *P. fruticosa* displayed the lowest range of genetic variability (Table 2).

Private alleles are alleles found only in one species and allow for judging genetic isolation. However, the analysis did not reveal any private alleles unique to any one species. For the three species (*P. griffithii* var. *tianshanica*, *P. erythrocarpa*, and *P. verrucosa*) that grow in the same habitat, this can be explained by possible natural hybridization between them. The absence of private alleles in *P*. *fruticosa* may be due to the small sample size, which decreased after excluding identical accessions from the analysis.

The AMOVA (analysis of molecular variance) showed that 81% of the variation was due to differences between the accessions within the respective species, and 19% of the variation was due to differences between the four cherry species (Figure 3).

The ϕ_PT_ value of 0.2, which is analogous to Fst, indicated moderate genetic differentiation between the species. The estimated number of migrants (Nm = 1.08) suggests ongoing gene flow between the species, likely preventing strong genetic divergence.

Nei’s genetic identity was determined based on the analysis of the pairwise population matrix, where higher values reflect a higher degree of genetic similarity between species. The highest value was observed when comparing *P. erythrocarpa* and *P. verrucosa* (0.50), confirming their high genetic similarity. The lowest genetic identity was observed between *P. griffithii* var. *tianshanica* and *P. fruticosa* (0.03), confirming their remarkable difference (Table 3).

### 2.3. Genetic Clustering and Phylogenetic Tree

The STRUCTURE results were analyzed using Structure Selector, which calculated Delta K according to the Evanno method to identify the most likely number of genetic clusters (K). As a result, 123 accessions of four wild *Prunus* species were grouped into three genetic clusters K = 3 (Delta K = 9.09) (Figure 4).

The STRUCTURE output reflects the species’ membership of each single accession in the three genetic clusters and is indicated by three different colors. Of the four species, the *P. fruticosa* accessions formed the most genetically distinct cluster (purple). The 25 *P. griffitii* var. *tianshanica* accessions collected in the Almaty region from a separate population, also formed a relatively distinct cluster (orange). In contrast, *P. erythrocarpa* and *P. verrucosa* were grouped into an admixed cluster and could not be clearly distinguished from each other (Figure 4). Despite the taxonomic classification of the four cherry species studied and despite the exclusion of putative hybrids at the beginning of the genetic analysis, the results suggest that the genetic separation between *P. erythrocarpa* and *P. verrucosa* is not very distinct. This explains why *P. erythrocarpa* and *P. verrucosa* remained genetically admixed and the program grouped them into the same cluster. To clarify these results, an additional principal coordinate analysis (PCoA) was performed using GenAlEx. This PCoA also revealed a genetic mixture of *P. erythrocarpa* and *P. verrucosa,* confirming the low genetic difference between the two cherry species (Figure 5). As for the STRUCTURE results, *P. fruticosa* formed the most genetically distinct cluster, and *P. griffithii* var. *tianshanica* also formed a relatively distinct cluster.

A phylogenetic tree was constructed using the Darwin program to evaluate the evolutionary relationships between different *Prunus* species. Unweighted neighbor-joining parameters with 10,000 bootstraps were used for tree construction. Additionally, this analysis included 57 accessions provided by the fruit gene bank of the Julius Kühn Institute (JKI) in Dresden-Pillnitz for comparison with species native to Kazakhstan and Uzbekistan. The *Prunus* accessions formed two main clusters (Figure 6). It should be noted that the bootstrap support for categorizing these main and sub-clusters was relatively low. Therefore, clustering should be interpreted with caution and considered as a tendency rather than a definitive phylogenetic separation. Conversely, the phylogenetic tree reflects the PCoA grouping (Figure 5), lending some confidence to the taxonomic grouping.

Cluster I separated into two subclusters. Subcluster I included the 14 *P. fruticosa* accessions (green lines) that were collected in Kazakhstan. Subcluster II included all the *Prunus* accessions in the JKI *Prunus* collections, containing 27 *Prunus* species (blue lines). The accession of *P. kurilensis* was the only exception, as it was grouped into an extra branch (black line). Interestingly, the two accessions of *P. fruticosa* from the JKI collection were also grouped into subcluster II rather than subcluster I, with the *P. fruticosa* accessions from Kazakhstan.

Cluster II was subdivided into two subclusters. Subcluster I contained 25 *P. griffithii* var. *tianshanica* accessions, growing separately in the Almaty region (pink lines). Subcluster II included the two species *P. verrucosa* (purple lines) and *P. erythrocarpa* (brown lines), which were collected in both Kazakhstan and Uzbekistan and grouped together, a result similar to the STRUCTURE output. Additionally, some *P. griffithii* var. *tianshanica* accessions were also grouped into this subcluster. However, a few accessions of *P. verrucosa* and *P. erythrocarpa* did not cluster with either subcluster I or subcluster II, forming a distinct branch and indicating a more distant genetic relationship with the other accessions of these species.

## 3. Discussion

This paper presents the results of a phenotypic and molecular genetic study of four wild species of *Prunus* indigenous to Kazakhstan and Uzbekistan.

### 3.1. Phenotypic Evaluation of Prunus Species

Phenotypic observations that were conducted during the expedition revealed that cherry bushes growing in protected national parks were generally taller and exhibited a higher percentage of fruiting compared to those in unprotected areas. For instance, smaller sized *P. fruticosa* bushes were observed in the Kostanay region, and similarly, *P. griffithii* var. *tianshanica* bushes were notably smaller in the Almaty region; both of these locations lack formal protection. In addition, a noticeable reduction in fruiting levels was recorded for bushes growing outside protected areas. These patterns of reduced plant size and lower fruit production are likely the result of human activities, including grazing and other forms of land use. Similar trends have been reported for *Prunus sibirica*, with human activities and environmental stressors having contributed to both habitat loss and population decline [18].

Identifying *Prunus* species using traditional morphological methods proved challenging, as some accessions showed morphological features that are typical for more than one species. For instance, *P. erythrocarpa* produces white tomentum on the abaxial side of the leaf, while *Prunus verrucosa* has brownish-gray branches covered with dry scales (warts). However, some of the studied accessions had both features simultaneously, making morphological classification challenging. This phenomenon was observed in populations where two or more species grew together, which may indicate the presence of hybridization between the species (e.g., populations in the Turkestan region, Sayram-Ugam National Park, Aksu-Zhabagly Nature Reserve, Kazakhstan, Tashkent region, and Ugam-Chatkal National Park, Uzbekistan, Table 1). Further results of our study also confirmed the limitations of morphological species identification. Four accessions morphologically identified as *P. griffithii* var. *tianshanica* in our study were assigned to another species after genetic analysis. The identification of *Prunus* species is also difficult because the phenotypic characteristics often change due to environmental conditions and the growth stage of the plant [19].

### 3.2. Genetic Diversity Parameters of the Prunus Species from Kazakhstan and Uzbekistan

SSR markers are reliable tools for assessing genetic diversity and population structure [19,20], which makes them advantageous for use in conservation and breeding programs. In this study, the genetic diversity and genetic structure of *Prunus* species were determined based on SSR markers. However, the 13 SSR markers used in this study could not distinguish between the following two *Prunus* species: *P. verrucosa* and *P. erythrocarpa*. This may be because the molecular markers used in this study were developed for *Prunus avium* varieties and are therefore not suitable for distinguishing wild *Prunus* species [16]. In the study by Chen et al. [21], the authors also highlighted the existing limitations of different markers (ISSR, RAPD, and RFLP-cpDNA) when applied to *Prunus pseudocerasus*. Further genetic analysis using chloroplast DNA markers, for example, could provide more precise data for differentiating the examined species. Further studies are planned in this context.

In order to identify existing hybrids in the accessions analyzed in our study, we employed the STRUCTURE analysis method with prior information on the population (POPINFO model). This approach was also used for studying indigenous wild apple species *Malus sylvestris* (Mill.) in Saxony, Germany [22]. As a result, 143 out of 163 accessions (87.7%) were identified as pure *Prunus* species, while the remaining 20 accessions were classified as hybrids (assignment probability <80%). We used a threshold of 80% assignment probability in the STRUCTURE analysis of each species to classify accessions as pure species. Accessions with probabilities below this threshold were classified as hybrids. Hybrids were found in three species (*P. griffithii* var. *tianshanica*, *P. erythrocarpa*, and *P. verrucosa*) but not in *P. fruticosa*. The highest percentage of hybrids was found in *P. griffithii* var. *tianshanica* (34.4%), which grew together with *P*. *verrucosa* and *P*. *erythrocarpa*. No hybrids were found in *P*. *griffithii* var. *tianshanica* accessions, growing separately in the Almaty region. Much smaller hybrid percentages were revealed in *P. erythrocarpa* (18.5%) and *P. verrucosa* (8.0%). These hybrid accessions were excluded from further analysis, since maintaining the genetic purity of the species is essential for preservation of this accession in the gene bank. Similar hybridization results were found in a study by Macková et al. on *P. fruticosa*, *P. cerasus*, and *P. avium,* in which 39.5% of the accessions in the studied populations were hybrids [23].

To identify genetically identical accessions, we used the ‘multilocus match analysis’ tool in GenAlex, which detects accessions with identical SSR profiles. In total, 26 out of the 143 accessions were identical in all four analyzed *Prunus* species. For further analysis, only one accession representative of each genotype was considered (six out of 26 accessions), while all duplicates were excluded. Identical genotypes were identified in all four species, with the highest number (54.8%) in *P. fruticosa*. This high frequency is probably related to *P. fruticosa* ability to reproduce vegetatively, particularly through the formation of root suckers [4].

Genetic diversity was high in all *Prunus* species analyzed in this study. The number of alleles per locus ranged from 8 (*P. fruticosa*) to a maximum of 28 (*P. verrucosa*). This level of genetic diversity is consistent with the data of previous studies of *Prunus* sp. [12,14,19,24,25,26,27].

The analysis of molecular variance (AMOVA) results showed that most of the genetic diversity was within the respective species (81%), while differences between species were moderate (19%). This indicates that, despite the *Prunus* species analyzed in our study being closely related and able to hybridize, there are still differences between them.

### 3.3. Genetic Clustering and Phylogenetic Tree

The subsequent analysis using the STRUCTURE program and PCoA further confirmed the differences between the four *Prunus* species analyzed in our study. However, although four taxonomic species were analyzed in our study, only three distinct genetic clusters were identified by using both STRUCTURE and PCoA analyses. While *P. fruticosa* and *P. griffithii* var. *tianshanica* formed clearly separated genetic groups, the genetic distance between *P. verrucosa* and *P. erythrocarpa* was low. This was likely due to the fact that several accessions could not be unambiguously assigned to either species. One possible explanation is that, co-occurring within the same populations, these species may have developed similar genetic traits under environmental influence, or alternatively, gene flow between them has led to genetic admixture. This observation highlights the potential for genetic exchange between *P. verrucosa* and *P. erythrocarpa* or may reflect the complexity of their taxonomic boundaries, which warrants further investigation.

In addition, a phylogenetic tree was constructed to assess the evolutionary relationships between *Prunus* species native to Kazakhstan and Uzbekistan and 27 different *Prunus* species from the JKI *Prunus* collection. The phylogenetic analysis produced results consistent with those of the STRUCTURE and PCoA analyses: both *P. fruticosa* and *P. griffithii* var. *tianshanica* accessions formed distinct clusters, whereas *P. verrucosa* and *P. erythrocarpa* were grouped together in a shared cluster, reflecting their close genetic relationship. Furthermore, all 57 accessions from the JKI collection, representing 27 *Prunus* species, clustered separately from the species native to Kazakhstan and Uzbekistan, highlighting a clear genetic distinction between these two groups. Interestingly, one *P. fruticosa* accession from the JKI collection did not cluster with the *P. fruticosa* accessions from Kazakhstan and Uzbekistan but instead grouped with the JKI *Prunus* collection cluster. This suggests that genetic distance among accessions is influenced not only by species identity but perhaps also by geographic origin.

## 4. Materials and Methods

### 4.1. Plant Material and Collection Sites

The objects of the study were four species of wild cherry: *P. fruticosa* Pall., *P. erythrocarpa* (Nevski) Gilli, *P. griffithii* var. *tianshanica* (Pojark.) Ingram, and *P. verrucosa* Franch. In 2024, expedition trips were organized to three regions of Kazakhstan, i.e., Almaty, Turkestan (Aksu-Zhabagly Nature Reserve and Sairam-Ugam National Park), and Kostanay, as well as to two regions of Uzbekistan, i.e., Tashkent (Ugam-Chatkal National Park) and Jizzakh (Zaamin National Park). The species were identified by botanists using descriptors [28]. Collection of plant material from protected areas was carried out in full compliance with national regulations. Official permits were obtained from the Ministry of Ecology and Natural Resources of the Republic of Kazakhstan.

In Kazakhstan, plant material (fruit and leaves) was collected from all four species, i.e., *P. griffithii* var. *tianshanica* (43 accessions), *P. erythrocarpa* (19 accessions), *P. verrucosa* (36 accessions), and *P. fruticosa* (31 accessions) (total 129). Accessions of the following three species were collected in Uzbekistan: *P. griffithii* var. *tianshanica* (6 accessions), *P. erythrocarpa* (14 accessions), and *P. verrucosa* (14 accessions) (total 34) (Figure 2 and Table 1). The map of the accession collection locations was generated using the following sources: https://ru.wikipedia.org/wiki/%D0%A4%D0%B0%D0%B9%D0%BB:Uzbekistan_location_map.svg (accessed on 10 February 2025) and https://ru.wikipedia.org/wiki/%D0%A4%D0%B0%D0%B9%D0%BB:Kazakhstan_location_map.svg (accessed on 10 February 2025). The coordinates of the growing areas were recorded in the WGS84 format using the eTREX^®^H Garmin GPS navigator. The latitude, longitude, altitude above sea level, and vegetation at the collection site with specification of the dominant species were recorded.

This study also included 57 accessions from the *Prunus* wild species collection of the fruit gene bank of the Institute for Breeding Research on Fruit Crops of the Julius Kühn Institute (JKI) in Dresden-Pillnitz (Table 4). These accessions comprised 27 wild *Prunus* species; the collection is designed as a permanent active field collection, a form of ex situ collection [29].

### 4.2. Phenotypic Assessment

For the phenotypic description of the 163 wild cherry accessions native to Kazakhstan and Uzbekistan, a list of descriptors recommended for *Prunus* was used [30,31]. A description of plants, leaves, and fruit was made. The following parameters were assessed: condition, vigor, height and width of the shrub, suckering tendency, yield efficiency, fruit size and shape, fruit juice color, fruit flesh color, eating quality, taste (sugar/acid ratio; organoleptic assessment was conducted to evaluate fruit taste and eating quality), firmness of flesh, flesh juiciness, skin cracking susceptibility, length of the fruit stalk, and leaves on the fruit stalk. For each plant, five leaves and from three to six fruit were photographed on a mapped sheet of paper to document their morphological characteristics. The leaf and fruit sizes were then automatically evaluated using a pixel-based segmentation method for each image, implemented in a Python 3.9.0 script (Appendix A).

### 4.3. DNA Extraction and SSR Analysis

The fresh leaves of 163 accessions collected during the expeditions were placed in plastic bags with filter paper in the presence of silica gel. Dried leaf material was stored at room temperature before DNA extraction. DNA extraction was carried out using the REDExtract-N-AmpTMPlant PCR Kit (Sigma-Aldrich, St. Louis, MO, USA). The quality of the extracted DNA was evaluated using Nanodrop (Thermo Scientific, Waltham, MA, USA).

Molecular genetic analysis was carried out for 13 microsatellite loci, i.e., CPPCT006, CPPCT022, EMPA002, EMPA003, EMPA017, EMPA026, EMPaS01, EMPaS02, EMPaS10, EMPaS12, EMPaS14, PS05C03, and UDP98-412, recommended for cherry species by the European Collaborative Programme for Genetic Resources (ECPGR) [16] (Table 5).

Multiplex PCR was performed using Type-It kit (Qiagen, Germany), with an initial denaturation at 95 °C for 5 min, followed by 40 cycles (denaturation at 95 °C for 30 s, annealing at 55 °C (48 °C for EMPaSO1 and PSO5CO3) for 1 min 30 s, and elongation at 72 °C for 1 min), with a final elongation step at 60 °C for 30 min.

Fragment lengths analysis was performed on a 3730XL DNA Analyzer (Applied Biosystems, Waltham, MA, USA) using the Software GeneMarker V2.6.4 (SoftGenetics LLC., State College, PA, USA). Some screenshots of the SSR analysis are present in the Appendix A.

### 4.4. Genetic Diversity Parameters

To confirm that the accessions were accurately identified as putative species, STRUCTURE software version 2.3.4 analyses were performed on 163 samples using prior information on the population (POPINFO model) [32]. To improve accuracy, the analysis was performed 5 times with K from 2 to 6, with parameters set to 50,000 burn-in periods and 50,000 Markov chain Monte Carlo repetitions. Accessions that showed <80% of probability of membership in the respective cherry species cluster were classified as hybrids and excluded from further analysis.

To identify identical genotypes, ‘multilocus matches analysis’ was performed using the software GENALEX ver. 6.5 [33,34]. This analysis was used to improve the accuracy of the genetic diversity estimates, as the presence of identical accessions may artificially inflate the frequency of certain genotypes.

Genetic diversity parameters such as mean number of alleles per locus (Na), effective number of alleles (Ne), observed heterozygosity (Ho), expected heterozygosity (He), and number of private alleles (PA) were estimated for 123 accessions collected in Kazakhstan and Uzbekistan using GENALEX version 6.5.

### 4.5. Genetic Structure and Phylogenetic Relationships Among the Prunus Species

To analyze the genetic structure of wild cherry species from Kazakhstan and Uzbekistan, STRUCTURE was run secondly without the POPINFO model [32]. The 123 cherry accessions from both countries were analyzed to identify potential subclusters. STRUCTURE was run with K values ranging from 2 to 6, performing five independent runs per K to assess the consistency of the results. The results were processed using STRUCTURE SELECTOR [35], and the most probable K value was determined using Evanno’s method (∆K) [36].

In addition, population genetic parameters including Shannon information index (I) and molecular variance (AMOVA) were also calculated to assess the genetic structure and differentiation among populations. To assess the degree of differentiation between genetic clusters, pairwise genetic distances (PhiPT values) and Nei genetic distance were calculated. Additionally, principal coordinate analysis (PCoA) was performed to identify genetic similarities and differences among species using GENALEX version 6.5.

For the phylogenetic analysis, a neighbor-joining tree was constructed using DARwin ver. 5 software based on the dissimilarity matrix obtained from the genetic data of the 123 wild cherry accessions from Kazakhstan and Uzbekistan and the 57 accession of the JKI wild *Prunus* species collection (total 180 accessions) [37]. Bootstrap analysis with 10,000 replications was used to assess the stability of clustering. A phylogenetic tree was constructed using the unweighted neighbor-joining method (UPGMA) in DARwin ver. 5. [38]. Visualization and further modification of the graphical representation of the tree was performed using Dendroscope ver. 2.7.4 [39].

## 5. Conclusions

In our study, morphological and molecular genetic analyses were conducted on four wild cherry species, i.e., *Prunus fruticosa* Pall., *Prunus erythrocarpa* (Nevski) Gilli, *Prunus griffithii* var. *tianshanica* (Pojark.) Ingram, and *Prunus verrucosa* (Franch.)*,* which are indigenous to Kazakhstan and Uzbekistan. The co-existence of these *Prunus* species in the same habitat combined with the possibility of natural hybridization made it difficult to identify the species based on morphological characteristics alone. However, genetic analysis based on 13 SSR markers enabled a clearer species assignment and allowed for the identification of hybrids or genetically identical accessions with the same SSR profiles. Cluster analysis revealed a distinct separation of *P. griffithii* var. *tianshanica* and *P. fruticosa*, while *P. erythrocarpa* and *P. verrucosa* showed a close genetic relationship, suggesting either ongoing gene flow or a recent divergence. Introgression, i.e., incorporation of genes from one species into the gene pool of another, was also observed, which is why the conservation of pure cherry species is threatened. Further research using additional molecular markers such as chloroplast DNA markers is recommended to better resolve the species boundaries and support the conservation of genetically pure accessions. These wild cherry species can be used in breeding—both as rootstocks and as donors of resistance genes (including resistance to drought, frost, diseases, and other stresses). Their gene pool is of great value for breeding adaptive varieties.

Additionally, several of the identified *Prunus* accessions were found in unprotected areas, where ongoing habitat degradation and population decline can be expected due to human activities. These findings highlight the critical importance of establishing and maintaining protected areas to safeguard wild plant populations and their reproductive potential.

## Figures and Tables

**Figure 1 plants-14-01676-f001:**
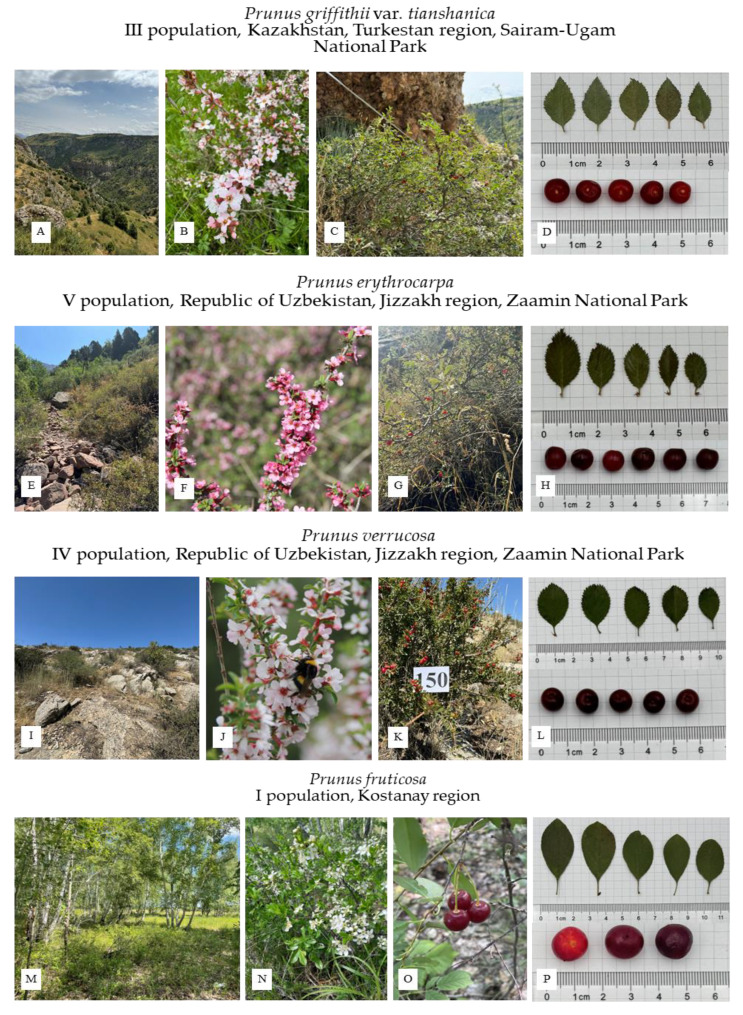
Plant material collection sites (**A**,**E**,**I**,**M**), flowering (**B**,**F**,**J**,**N**), fruiting (**C**,**G**,**K**,**O**) leaves, and fruits (**D**,**H**,**L**,**P**) of four cherry species.

**Figure 2 plants-14-01676-f002:**
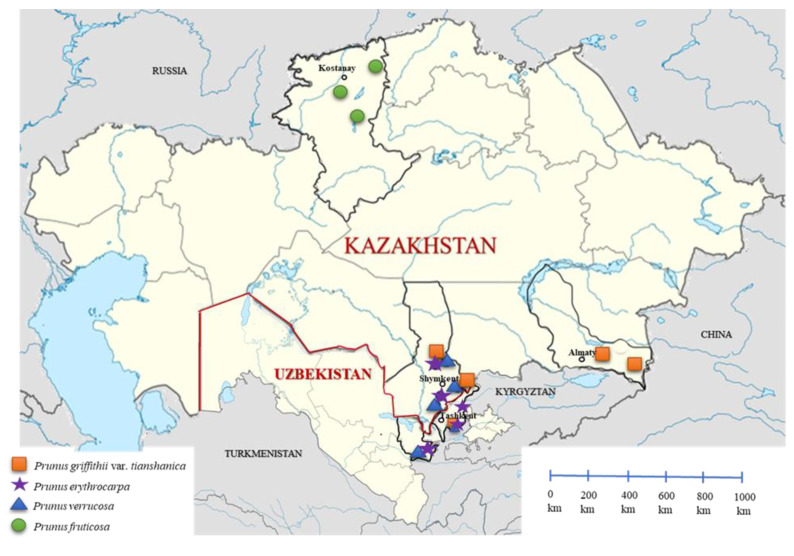
Collection sites of four cherry species in natural populations in Kazakhstan and Uzbekistan.

**Figure 3 plants-14-01676-f003:**
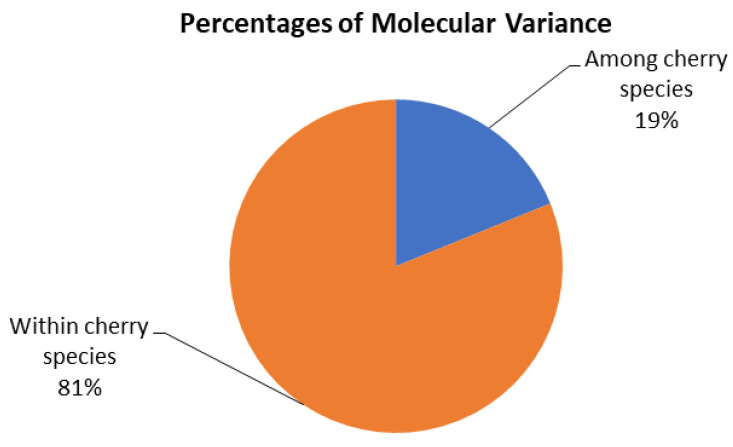
Molecular variance between and within wild *Prunus* species.

**Figure 4 plants-14-01676-f004:**
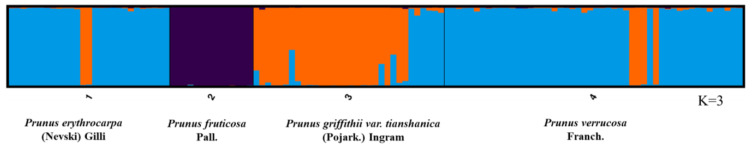
Clusters were derived from structural analysis of 123 cherry accessions based on 13 SSR markers. Each column corresponds to each individual divided into three genetic clusters (K = 3). Each color represents the estimated proportion of membership in the three genetic clusters (cluster 1 = blue; cluster 2 = purple, cluster 3 = orange).

**Figure 5 plants-14-01676-f005:**
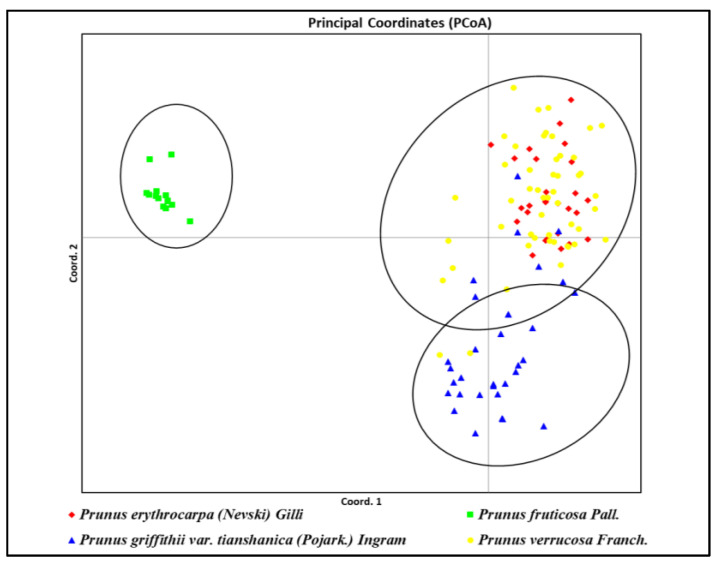
Principal coordinate analysis (PCoA) of pairwise distances among *Prunus erythrocarpa*, *Prunus fruticosa*, *Prunus griffithii* var. *tianshanica,* and *Prunus verrucosa*.

**Figure 6 plants-14-01676-f006:**
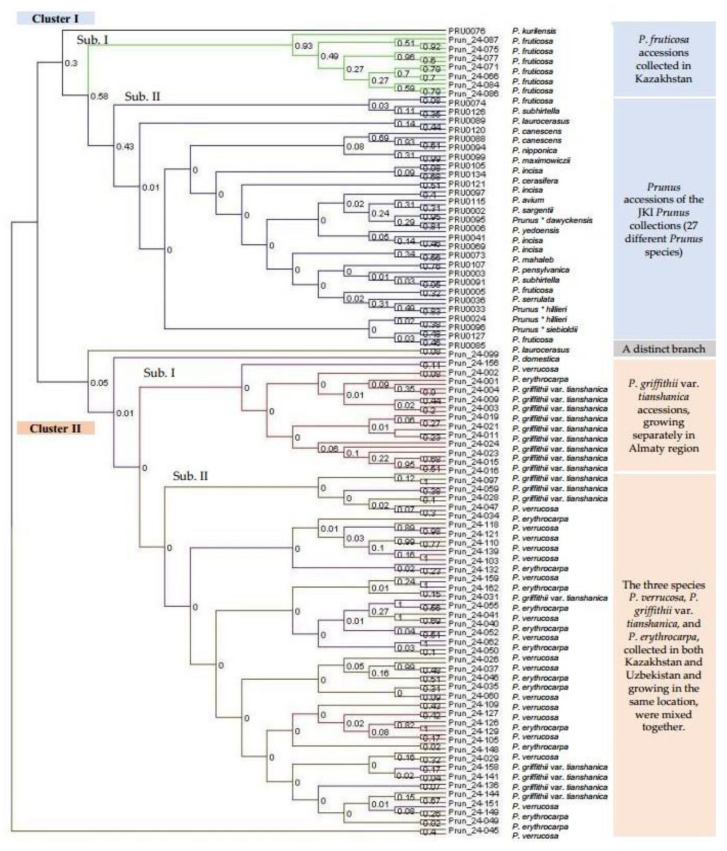
Dendrogram obtained by processing data from 123 accessions collected in Kazakhstan and Uzbekistan and 57 accessions from the fruit gene bank of the Institute for Breeding Research on Fruit Crops of the Julius Kühn Institute (JKI) in Dresden-Pillnitz (total 180 accessions). The *Prunus* species accessions are labeled by different colors: *Prunus verrucosa* is marked in brown, *Prunus griffithii* var. *tianshanica* is marked in red, *Prunus erythrocarpa* is marked in purple, *Prunus fruticosa* is marked in green, and the accessions from the German fruit bank collection are marked in blue. A few accessions of *P. verrucosa* and *P. erythrocarpa* did not cluster into either subclusters I or II but formed a distinct branch.

**Table 1 plants-14-01676-t001:** Origin of *Prunus* accessions from Kazakhstan and Uzbekistan.

Species	Location	Population	GPS Coordinates (WGS84)			Number of Accessions
			Latitude	Longitude	Altitude Above Sea Level, m	
*Prunus griffithii* var. *tianshanica*	Kazakhstan, Almaty region	Tian 1	43°27′09.41″	78°39′16.94″	1117–1171	10
		Tian 2	43°21′01.28″	79°37′10.40″	1438–1488	15
	Kazakhstan, Turkestan region, Sairam-Ugam National Park	Tian 3	43°04′14.19″	69°54′01.76″	695–730	9
		Tian 4	42°57′19.06″	70°02′58.11″	836–864	7
	Uzbekistan, Tashkent region, Ugam-Chatkal National Park	Tian 5	41°31′05.16″	70°01′05.03″	1689–1798	4
Total						45
*Prunus erythrocarpa*	Kazakhstan, Turkestan region, Aksu-Zhabagly Nature Reserve	Eryth 1	42°24′48.36″	70°28′14.82″	1252–1534	16
	Kazakhstan, Turkestan region, Sairam-Ugam National Park	Eryth 2	44°11′21.00″	70°31′34.80″	716	3
	Uzbekistan, Tashkent region, Ugam-Chatkal National Park	Eryth 3	41°36′00.00″	70°06′16.00″	1690–1756	9
		Eryth 4	41°13′46.00″	70°14′40.00″	1543–2069	5
	Uzbekistan, Jizzakh region, Zaamin National Park	Eryth 5	39°52′38.00″	68°28′27.00″	2060	1
Total						34
*Prunus verrucosa*	Kazakhstan, Turkestan region, Aksu-Zhabagly Nature Reserve	Ver 1	42°19′49.86″	70°22′22.26″	1123–1598	24
	Kazakhstan, Turkestan region, Sairam-Ugam National Park	Ver 2	42°40′07.21″	70°13′45.82″	848–854	5
		Ver 3	43°05′45.43″	69°55′44.34″	715–733	9
	Uzbekistan, Tashkent region, Ugam-Chatkal National Park	Ver 4	41°10′03.00″	70°14′13.00″	1572	2
	Uzbekistan, Jizzakh region, Zaamin National Park	Ver 5	39°44′24.00″	68°38′03.00″	2029–2233	13
Total						34
*Prunus fruticosa*	Kazakhstan, Kostanay region	Frut 1	53°02′14.40″	63°40’35.22″	187–199	11
		Frut 2	53°17’18.00″	64°12’31.80″	190–202	7
		Frut 3	52°26’35.40″	64°18’30.00″	153–169	13
Total						31
Total						163

**Table 2 plants-14-01676-t002:** Genetic diversity parameters of *Prunus* species indigenous to Kazakhstan and Uzbekistan.

Species	N	Na	Ne	I	Ho	He	uHe	F
*Prunus erythrocarpa*	15.19	8.96	5.82	1.63	0.52	0.66	0.70	0.21
*Prunus fruticosa*	8.12	2.19	1.71	0.54	0.42	0.30	0.32	−0.41
*Prunus griffithii* var. *tianshanica*	17.73	7.81	4.57	1.55	0.51	0.68	0.71	0.27
*Prunus verrucosa*	28.62	14.04	8.53	2.10	0.57	0.80	0.84	0.27
Mean	17.41	8.25	5.16	1.46	0.51	0.61	0.64	0.15
SE	1.45	0.71	0.41	0.09	0.04	0.03	0.03	0.05

*N:* number of alleles; Na: number of different alleles; Ne: number of effective alleles (=1/(∑ pi2)); I: Shannon’s information index = −1 ∗ Sum (pi ∗ Ln (pi)); Ho: observed heterozygosity (=number of heterozygotes/N); He: expected heterozygosity (=1 − ∑ pi2); uHe: unbiased expected heterozygosity = (2N/(2N−1)) ∗ He; F = fixation index = (He − Ho)/He = 1 − (Ho/He).

**Table 3 plants-14-01676-t003:** The analysis of pairwise population matrix of Nei genetic identity between *Prunus* species.

*Prunus erythrocarpa*	*Prunus fruticosa*	*Prunus griffithii* var. *tianshanica*	*Prunus verrucosa*	
1.00				*Prunus erythrocarpa*
0.09	1.00			*Prunus fruticosa*
0.44	0.03	1.00		*Prunus griffithii* var. *tianshanica*
0.50	0.09	0.46	1.00	*Prunus verrucosa*

**Table 4 plants-14-01676-t004:** Wild *Prunus* species accessions from the collection of the fruit gene bank of the Institute for Breeding Research on Fruit Crops of the Julius Kühn Institute (JKI) in Dresden-Pillnitz.

Species	Number of Accessions in the JKI Gene Bank Collection
*Prunus × cistena*	1
*Prunus × dawyckensis*	3
*Prunus × hillieri*	3
*Prunus × siebioldii*	1
*Prunus avium*	2
*Prunus brigantina*	1
*Prunus canescens*	4
*Prunus cerasifera*	2
*Prunus domestica*	3
*Prunus fruticosa*	2
*Prunus incisa*	6
*Prunus kurilensis*	2
*Prunus laurocerasus*	2
*Prunus maackii*	1
*Prunus mahaleb*	3
*Prunus maximowiczii*	2
*Prunus mollis*	1
*Prunus nipponica*	2
*Prunus padus*	1
*Prunus pensylvanica*	1
*Prunus sargentii*	2
*Prunus serrula*	1
*Prunus serrulata*	4
*Prunus subhirtella*	2
*Prunus tomentosa*	2
*Prunus virginiana*	1
*Prunus yedoensis*	2
Accessions total	57

**Table 5 plants-14-01676-t005:** SSR markers used for *Prunus* species genotyping.

Primer Mix	Primer	Dye	Size Range	PCR Annealing Temp.
Primer Mix 1	EMPA002	At532	98–133	55 °C
	EMPA003	At550	161–181	55 °C
	CPPCT006	FAM	172–205	55 °C
	EMPaSO2	At565	130–187	55 °C
Primer Mix 3	CPPCTO22	FAM	219–235	55 °C
	EMPaS14	At565	164–212	55 °C
	UDP98-412	At532	83–155	55 °C
	EMPaSO1	At532	199–250	48 °C
	EMPaS12	FAM	100–150	55 °C
	EMPaS10	At565	135–207	55 °C
Primer Mix 4	EMPA017	FAM	218–244	55 °C
	EMPA026	At532	179–258	55 °C
	PSO5CO3	At550	105–168	48 °C

## Data Availability

The presented research results are included in the article and Appendix A. Further inquiries can be directed to the corresponding authors.

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
