# Peer review of "Genetic Diversity and Phenotypic Variation of Indigenous Wild Cherry Species in Kazakhstan and Uzbekistan"

_plants, 2025, doi:10.3390/plants14111676_

Round 1
Reviewer 1 Report
Comments and Suggestions for Authors
Dear Authors,
I thoroughly read your manuscript, and I suggest that you consider the following points for improving the quality of your manuscript.
Abstract
- Percentages are inconsistent and potentially confusing (e.g., mixing exact numbers with percentages like “143 accessions (87.7%)” and then “34.4%, 18.5%, and, 8.0%” without clear referents).
- The abstract jumps between morphology, genetic analysis, and specific findings—reorganizing into a logical flow (objectives → methods → main findings → significance) would improve readability.
- The last sentence could be made more impactful by clearly stating the specific applications for breeding or conservation rather than a general “highlight their potential.”
Introduction
- For flow of the introduction, consider separating into more focused paragraphs: (1) global importance of cherry, (2) wild species in the region, (3) their breeding potential, and (4) research gaps.
- Consider clarifying which source supports which specific claim, especially in taxonomic and regional data.
- “To our knowledge, this is the first work...” → could be made stronger and more formal: “This study represents the first comprehensive molecular and phenotypic analysis of these four wild cherry species from Kazakhstan and Uzbekistan.”
- Please provide a concrete hypothesis for this study at the end of introduction.
Results
- Reduce botanical species lists unless directly relevant to the study
- Be careful of Overusing passive voice: E.g., “accessions were collected” can sometimes be more actively phrased for engagement.
- Altitudes sometimes appear without context; consider standardizing format and including elevation range once per species.
- Leaf area, height, and fruit dimensions are important but lack statistical summary (mean ± SD or range with N).
- Fruit taste description (“sour,” “sweet”) is too subjective — consider whether these are quantified or based on observation.
- Improve transitions showing how morphological data connects with molecular reclassification.
Discussion
- Rephrase redundant or awkward constructions (e.g., “were not able to accurately distinguish” → “could not distinguish”).
- Use subheadings (e.g., “Phenotypic Observations”, “Molecular Identification”, “Genetic Diversity”, “Hybridization”) to improve flow and readability.
- Avoid unnecessary repetition (e.g., overuse of "highlights the limitations of morphological identification").
- Provide brief details or references to the limitations of SSR markers specifically for wild Prunus species.
- Clarify STRUCTURE analysis results (e.g., “<80” assignment threshold should be better explained).
- Ensure citation style consistency and update citation numbers if needed.
- Use consistent scientific naming conventions (e.g., Prunus griffithii tianshanica instead of inconsistent “griffiithii”).
Materials and methods
- GPS description: “eTREX®H Garmin Montana 750i GPS navigator” — eTREX and Montana are two different models; clarify which one was used.
- In Image analysis method, the "pixel-based segmentation method" in Python is mentioned but lacks sufficient detail or a proper reference.
- In DNA kit and tools, Manufacturer info is inconsistently formatted (e.g., "The Thermo Scientific™" vs. "Qiagen, Germany").
- Add sample sizes clearly at the beginning of each relevant subsection.
Conclusion
- “Introgression” and “genetically identical accessions” may need clarification for broader audiences or general readers.
- The suggestion for using “additional molecular markers” could be more specific (e.g., SNPs, plastid markers).
The quality of English in this manuscript would benefit from thorough proofreading by a native speaker.
Reviewer 2 Report
Comments and Suggestions for Authors
The author should specify who identified the four cherry species , and also indicate that the sample collection method was reasonable and legal.
There is one concern if the author's sampling strategy was reasonable. there are different number in these sites, why , does the number cover all the population samples? the author should explain the reasons.
Reviewer 3 Report
Comments and Suggestions for Authors
The manuscript presents a comprehensive study on the phenotypic characteristics, genetic diversity, and genetic structure of four wild cherry species collected from various regions of Kazakhstan and Uzbekistan. The research is well-structured and provides valuable insights into the genetic diversity and potential breeding and conservation applications of these species. The findings are significant for breeding programs aimed at improving traits such as drought and frost resistance in cherry species.
The following are two suggestions for the author to consider:
(1)It is recommended to add some images output from the 3730XL DNA Analyzer.
(2)Figure 6 should be modified to label accessions with their material numbers instead of Latin names, include Coefficient values, indicate groupings, and present Latin names in italics.
Overall, this manuscript provides a valuable contribution to the understanding of wild cherry species' genetic diversity and structure.
Round 2
Reviewer 1 Report
Comments and Suggestions for Authors
The authors addressed all the comments. However, the English quality of the manuscript still needs improvement. Please check the flow and coherence of the text with a native colleague.
Author Response
Comments and Suggestions for Authors 1: The authors addressed all the comments. However, the English quality of the manuscript still needs improvement. Please check the flow and coherence of the text with a native colleague.
Response 1: We thank the reviewer for his valuable comments. Our manuscript was reviewed by a native speaker. We also worked on improving the overall flow and coherence of the text accordingly. All corresponding revisions have been incorporated into the revised manuscript and are highlighted using track changes in the re-submitted files.